# Understanding the role of diabetes in the osteoarthritis disease and treatment process: a study protocol for the Swedish Osteoarthritis and Diabetes (SOAD) cohort

Andrea Dell'Isola [1,2] Johanna Vinblad,[3,4] Stefan Lohmander,[1] Ann-Marie Svensson,[5,6] Aleksandra Turkiewicz,[2] Stefan Franzén,[7,8] Emma Nauclér,[3] A W-Dahl,[9,10] Allan Abbott,[11] L Dahlberg,[1] Ola Rolfson,[3,4] Martin Englund[2]

AD and JV are joint first authors. OR and ME are joint senior authors.

For numbered affiliations see end of article.

**Correspondence to**
Dr Andrea Dell'Isola; andrea.dellisola@med.lu.se

## ABSTRACT

**Introduction** Osteoarthritis (OA) is the most common form of arthritis and a leading cause of disability worldwide. Metabolic comorbidities such as type II diabetes occur with a higher rate in people with OA than in the general population. Several factors including obesity, hyperglycaemia toxicity and physical inactivity have been suggested as potential links between diabetes and OA, and have been shown to negatively impact patients' health and quality of life. However, little is known on the role of diabetes in determining the outcome of non-surgical and surgical management of OA, and at the same time, how different OA interventions may affect diabetes control. Thus, the overall aim of this project is to explore (1) the impact of diabetes on the outcome of non-surgical and surgical OA treatments and (2) the impact of non-surgical and surgical OA treatments on diabetes control.

**Methods and analysis** The study cohort is based on prospectively ascertained register data on a national level in Sweden. Data from OA patients who received a first-line non-surgical intervention and are registered in the National Quality Register for Better Management of Patients with Osteoarthritis will be merged with data from the Swedish Knee and Hip Arthroplasty Registers and the National Diabetes Register. Additional variables regarding patients' use of prescribed drugs, comorbidities, socioeconomic status and cause of death will be obtained through other national health and population data registers. The linkage will be performed on an individual level using unique personal identity numbers.

**Ethics and dissemination** This study received ethical approval (2019-02570) from the Swedish Ethical Review Authority. Results from this cohort will be submitted to peer-reviewed scientific journals and reported at the leading national and international meetings in the field.

### Strengths and limitations of this study

► This study will use a large nationwide population-based cohort based on data from national quality registers with high coverage and completeness to explore the relationship between diabetes and osteoarthritis (OA) and their related care process.

► We will include data regarding both non-surgical and surgical treatments for patients with OA giving the possibility to capture the influence of diabetes across the whole spectrum of OA treatments.

► We will include covariate information from several national registers that will allow to account for potential confounders and effect modifiers.

► A limitation of register-based studies is that the variables available and the characteristics of the treatments provided are predetermined, that is, it is not possible to add covariates, exposures or outcomes (not in the registers) or to modify the interventions that have been given.

► People included in the National Quality Register for Better Management of Patients with Osteoarthritis Register, the Swedish Hip Arthroplasty Register, the Swedish Knee Arthroplasty Register received an intervention due to OA. Due to the complexity of the OA disease, treatments are individualised based on patient's and disease characteristics, which implies that selection bias and confounding by indication may bias our estimates.

## INTRODUCTION

Osteoarthritis (OA) is the most common form of arthritis and affect mainly the knee and the hip joint.[1] In Sweden, more than 25% of the population aged >45 years is estimated to suffer from OA-related pain symptoms and associated physical activity restrictions.[2] The average annual cost for a person affected by OA is reported to exceed €2000, while the total European expense directly attributable to OA is estimated to be as high as €700 billion.[3]

In addition to the already huge health and societal burden of OA, recent studies suggest that OA patients are twice as likely to have

comorbidities compared with controls of the same age, indicating that the co-occurrence of multiple conditions in OA patients is the norm rather than the exception.[4] For instance, based on data showing a higher incidence of knee OA in overweight patients with metabolic disorders, a metabolic OA phenotype has been hypothesised.[5–7] Among the metabolic disorders, diabetes seems to play a central role due to its high prevalence and the toxic effect that hyperglycaemia has on the cartilage and its cells, and to the motor and sensory system through peripheral neuropathy.[8–10]

According to data from the Swedish National Diabetes Register (NDR), approximately 5% of the Swedish population has diabetes, with type II diabetes accounting for approximately 90% of the cases.[11] Persons with diabetes have a higher risk of developing cardiovascular diseases and have a twofold to fivefold increased risk of mortality compared with the general population.[12]

In OA patients the prevalence of diabetes has been reported to be nearly three times higher than in the general population.[7] Obesity is a shared risk factor for OA of the knee and the hip and diabetes and can partially explain the association between these diseases.[13 14] In addition to the mechanical overload caused by the excess weight, adipocytes release cytokines into the bloodstream promoting chronic low-grade inflammation and activating proteolytic enzymes which can trigger matrix degradation and initiate OA. At the same time, adipose-induced low-grade inflammation influences the metabolic dysregulation underlying several metabolic disorders, for example, diabetes type II.[14]

Once diabetes is initiated, it further promotes cartilage degeneration and joint inflammation causing enrichment of advanced glycation end products and matrix stiffening preventing optimal cushioning of the joint.[7 15] This process leads to a worsening in OA symptoms promoting physical inactivity and weight gain and creating a vicious cycle that maintains the metabolic dysregulation and increases joint symptoms.[13 16–18]

The evidence-based first-line management for people with hip and knee OA includes education and exercise which are recommended regardless of OA disease severity, and weight loss for those overweight.[19] Metabolic comorbidities may have a significant impact on the treatment, partially explaining the lack of response experienced by some patients.

Replacement of the knee and of the hip is an effective treatment for patients with severe OA who do not sufficiently improve after non-surgical management.[20] Due to the rising prevalence of OA and the growing demand for this procedure, the number of hip and knee replacements has dramatically increased. In Sweden (total population 10 million), 14 700 primary total hip replacements (THRs) and nearly 14 000 total knee replacements (TKRs) were performed in 2017 with OA as indication. These figures account for 81% and 97% of the annual hip and knee replacements, respectively, and translate in an annual incidence of nearly 150 procedures per 100 000 persons for both THR and TKR.[21 22]

Considering the association between diabetes and OA, surprisingly little is known regarding the influence that diabetes has on the outcome of OA treatments (both non-surgical and surgical).[23 24] In addition, no evidence exists regarding the effect that OA treatments (both non-surgical and surgical) may have on diabetes control (both for types I and II). Thus, merging data from multiple Swedish registers will allow us to follow patients with knee and hip OA through the progress of their disease to understand how diabetes influences the OA disease process. This study cohort is created to increase knowledge of the influence that diabetes has on the outcomes of OA patients who have received non-surgical and/or surgical treatments for hip and knee OA, and the influence that hip and knee OA and its treatments have on the diabetes control.

## METHODS AND ANALYSIS
### Research questions
In order to understand how the coexistence of OA of the hip or of the knee and diabetes influences the treatment effects in these diseases, a series of research questions have been posed. The research questions cover two main thematic areas: (1) the impact of diabetes on the outcome of non-surgical and surgical OA treatments and (2) the impact of non-surgical and surgical OA treatments on diabetes control (consideration to type of diabetes (I or II) will be taken).

### Area 1
1. What is the prevalence of diabetes in people with OA undergoing a non-surgical intervention?
2. Is the presence of diabetes, diabetes-related factors (eg, type of diabetes, diabetes-related medication, blood pressure, haemoglobinsubunit alpha 1c (HbA1c)) associated with OA severity (eg, pain intensity, pain frequency, walking difficulties) of people with OA undergoing a self-management non-surgical intervention?
3. Is the presence of diabetes and diabetes-related factors associated with the outcomes of a self-management non-surgical intervention for people with OA (eg, change in pain levels, pain frequency, walking difficulties)?
4. Is the presence of diabetes and diabetes-related factors associated with the risk of joint replacement in people with OA who underwent a self-management non-surgical intervention?
5. What is the incidence of re-operations and other adverse events such as thromboembolism, cardiovascular events and mortality following primary THR or TKR due to OA in people with or without diabetes?
6. What diabetes-related factors are associated with the risk of re-operation and other adverse events following primary THR or TKR among person with diabetes?

## Area 2

1. How does a self-management non-surgical intervention for OA influence diabetes (type I vs type II) control (eg, change in diabetes drug intake after the intervention, change in HbA1c levels after the intervention) compared with people with diabetes who had not taken part in the intervention?
2. How does primary THR or TKR influence the diabetes control compared with comparable persons with diabetes but with no history of hip or knee arthroplasty?

3. What diabetes-related risk factors are associated with diabetes control following primary THR or TKR due to OA?

### Main exposures and outcomes

The exposure and outcome measures are described in table 1. Potential confounding factors for main analysis and disease subanalysis are described with examples.

### The Swedish OA and diabetes cohort

This nationwide observational study cohort (Swedish OA and diabetes (SOAD)) will be based on prospectively

**Table 1** Exposure and outcome for the study populations and examples of confounders and effect modifiers for the study analyses

| Population | Exposure | Outcome | Example confounders and effect modifiers |
|---|---|---|---|
| **People with OA** | | | |
| Undergoing self-management treatment | Diabetes | ▶ Pain intensity.<br>▶ Pain frequency.<br>▶ Walking difficulties.<br>▶ Quality of life.<br>▶ Use of pain medications.<br>▶ Sick leave. | ▶ Patient's characteristics (age, sex, BMI, smoking).<br>▶ Type of diabetes.<br>▶ Diabetes medications.<br>▶ Diabetes severity (eg, HbA1c, blood pressure, cholesterol levels, albuminuria).<br>▶ Diabetes disease duration (age at diagnosis, duration of disease).<br>▶ Diabetes-related complications (eg, ocular bottom changes, kidney disease, neuropathy).<br>▶ Cardiovascular comorbidities.<br>▶ Physical activity.<br>▶ Weight change. |
| Undergoing surgical OA treatment | Diabetes | ▶ Implant survival.<br>▶ Re-operation within 2 years.<br>▶ Change in patient-reported outcome measures.<br>▶ Adverse events (eg, cardiovascular events).<br>▶ Mortality. | ▶ Patient's characteristics (age, sex, BMI, smoking).<br>▶ Type of diabetes.<br>▶ Diabetes medications.<br>▶ Diabetes severity (eg, HbA1c, blood pressure, cholesterol levels, albuminuria).<br>▶ Disease progression (age at diagnosis, duration of disease).<br>▶ Diabetes-related complications (eg, ocular bottom changes, kidney disease, neuropathy).<br>▶ Cardiovascular comorbidities.<br>▶ Weight change. |
| **Diabetes** | | | |
| | Non-surgical OA treatment of hip and knee | ▶ Diabetes medications (diabetes, blood sugar, lipid and blood pressure lowering).<br>▶ Diabetes severity (eg, HbA1, blood pressure, cholesterol levels, albuminuria).<br>▶ Diabetes-related complications (eg, ocular bottom changes, kidney disease, neuropathy). | ▶ Patient's characteristics (age, sex, BMI, smoking).<br>▶ OA severity (pain intensity, pain frequency, walking difficulties, quality of life, pain medications, sick leave).<br>▶ Type of diabetes.<br>▶ Cardiovascular comorbidities.<br>▶ Physical activity.<br>▶ Weight change. |
| | Surgical OA treatment of hip and knee | ▶ Diabetes medications (diabetes, blood sugar, lipid and blood pressure lowering).<br>▶ Diabetes severity (eg, HbA1c, blood pressure, cholesterol levels, albuminuria).<br>▶ Diabetes-related complications (eg, ocular bottom changes, kidney disease, neuropathy). | ▶ Patient's characteristics (age, sex, Charnley classification, BMI).<br>▶ Type of diabetes.<br>▶ Surgical technique.<br>▶ Implant characteristics.<br>▶ Cardiovascular comorbidities.<br>▶ Weight change. |

BMI, body mass index; HbA1, haemoglobin subunit alpha 1; HbA1c, haemoglobin subunit alpha 1c; OA, osteoarthritis.

obtained individual-level data from four main sources: the National Quality Register for Better Management of Patients with Osteoarthritis (BOA) Register, the Swedish Hip Arthroplasty Register (SHAR), the Swedish Knee Arthroplasty Register (SKAR) and the NDR. Data starting from the year of each register establishment will be merged using the unique personal identity number (PIN) issued to all legal residents in Sweden. Additional variables regarding patients' use of prescribed drugs, comorbidities, cause of death and socioeconomic information will be obtained through the following population-based registers:

▶ The Swedish Prescribed Drug Register held by the National Board of Health and Welfare.
▶ The National Patient Register held by the National Board of Health and Welfare; information regarding in-hospital diagnoses and outpatient specialist care diagnoses, for example, interventions, adverse events such as thromboembolism or other comorbid conditions.
▶ Swedish Cancer Register, The National Board of Health and Welfare.
▶ The Cause of Death Register held by the National Board of Health and Welfare
▶ Longitudinal integration database for health insurance and labour market studies (LISA) held by Statistics Sweden for data such as marital status, educational level and country of origin.

## Data sources

BOA: The BOA register was started in 2008 and currently includes more than 100 000 individuals with OA who have registered for an evidence-based self-management programme. These patients sought treatment for knee and/or hip pain in primary healthcare in Sweden and were referred for standardised core treatment (education and supervised exercises) after a confirmed clinical/radiographic OA diagnosis in accordance with the recommendations for OA diagnosis from the Swedish National Board of Health and Welfare.[25] These guidelines are in line with internationally accepted diagnostic criteria, and according to the guidelines, radiographic examination should only be used in uncertain cases, if the patient is not responding to treatment or when a surgical intervention is planned.[26 27] BOA offers all the patients two education sessions focusing on the pathophysiology of OA and the benefit of exercise which are mandatory for participating in the second (exercise) session of the programme. A third, optional, session held by a trained OA communicator (a patient with OA who previously participated in BOA) is offered to provide a patient's perspective on OA self-management and to teach about the lived experience with this condition, as well as his or her personal experience of non-surgical interventions. After the education, participants can take part in the exercise phase of BOA which consists of a face-to-face session with a physiotherapist. In this session, the patients receive a personalised intervention programme and the necessary instructions

to perform it independently at home. Thereafter, participants are given the possibility to perform their exercise programme on their own or to participate in up to 12 supervised group exercise session with a physiotherapist provided two times per week for 6 weeks. Thus, the register contains two separate cohorts that performed, in addition to the education sessions, either home exercise or supervised exercise. The register has a data completeness of almost 90% and the BOA participants have answered validated and patient-relevant sociodemographic and outcome questionnaires at baseline, after the interventions (2–5 months) and at 1 year (12–15 months) (table 2).

SHAR: Started in 1979, SHAR registers primary hip replacement operations and re-operations in Sweden, including individual patient data, surgical technique and type of implant used. Since 2002 patient-reported measures such as joint pain, Health Related Quality of Life (HRQoL) and satisfaction with treatment have also been collected before surgery and 1, 6 and 10 years postoperatively. The register encompasses 318 000 primary THRs due to OA and 61 500 re-operations after THRs where OA was the main reason for the primary surgery (at the end of 2018). The register has overall data completeness of 98.5% (2016) including all indications for THRs (table 3).

SKAR: The SKAR is a Swedish National Quality Register founded in 1975. The register collects individual patient data, surgical technique and type of implant used for patients who undergo knee replacement. The SKAR also collects information on re-operations/revision surgery. SKAR has completeness of 98.1% (2016) and has registered almost 270 000 primary knee replacements due to OA and more than 21 400 revisions at the end of 2018 (table 4).

NDR: NDR has been a Swedish National Quality Register since 1996 and collects data on clinical characteristics, risk factors, laboratory analyses, complications of diabetes and medications for patients 18 years of age or older with a diagnosis of diabetes (table 5). The completeness is 96.5% (2017) and the register has 750 004 (2017) unique individuals in their database. More than 95% of all individuals with type I diabetes mellitus (T1DM) and 90% of individuals with type II diabetes mellitus (T2DM) in Sweden are included in the NDR.

## Data linkage

PIN: In Sweden, all legal residents are registered with a unique PIN that provides information on the date of birth and sex. Swedish law requires all documentation regarding healthcare contacts to be registered using the patient's PIN.[28] The PIN is also used for registration of data for statistics such as national population-based registers and healthcare quality registers.[29 30] The system allows for linkage of data at an individual level between the different registers in Sweden with the possibility of creating merged research databases for epidemiological research on large populations, after the relevant ethical

**Table 2** Description of the single variables collected from the BOA register

| BOA register | | Baseline | Evaluation 3 months | Evaluation 12 months |
|---|---|---|---|---|
| **Variable category** | **Variable** | | | |
| Date | Date of visits | x | x | x |
| *Patient-reported measures* | Age, sex, weight, height | x | | |
| | Smoking | x | | |
| | Most affected joint (hip, knee or hand) | x | x | x |
| | Other affected joints | x | x | x |
| | Fear avoidance | x | x | x |
| | Request for surgery | x | x | x |
| | Walking difficulties | x | x | x |
| Physical activity level | Duration of physical training* | x | x | x |
| | Duration of physical activity† | x | x | x |
| Satisfaction | Satisfaction with treatment | | x | x |
| Musculoskeletal comorbidity | Charnley class‡ | x | x | x |
| Pain | Pain severity‡ NRS | x | x | x |
| | Pain frequency | x | x | x |
| Generic | EQ-5D | x | x | x |
| Self-efficacy | Arthritis self-efficacy scale | x | x | x |
| *Physiotherapist-reported measures* | Earlier radiography/MRI/surgery in the most affected or the contralateral joint | x | | |
| | Earlier treatments (including physiotherapy/ adapted training/information on weight reduction/pharmaceuticals) | x | | |
| | Waiting list for surgery | x | x | |
| | Use of medications for OA | x | x | |
| Follow-up | Radiography/MRI/surgery in the most affected or the contralateral joint since last evaluation | | x | |
| | Compliance with intervention | | x | |

*Answering to the question: 'During a regular week, how much time do you spend exercising on a level that makes you short winded, for example running, fitness class, or ball games?' graded on categorical scale from '0' to 'more than 120 min'.
†Answering to the question: 'During a regular week, how much time are you physically active in ways that are not exercise, for example walks, bicycling, or gardening?' graded on categorical scale from '0' to 'more than 300 min'.[25]
‡Charnley class: classifications of musculoskeletal impairment. Class A corresponds with unilateral hip or knee OA; class B bilateral hip or knee OA and class C indicates multiple joint OA or some other condition that inhibits the patient's ability to walk.
§Answering to: 'Select the box that corresponds to your average pain from your most affected joint the last week'.
BOA, Better Management of Patients with Osteoarthritis; NRS, Numeric Rating Scale; OA, osteoarthritis.

approval has been obtained. Data linkage for the current study will include all the data available in BOA, SHPR, SKAR and NDR and it will start from the first time point available in the registers. The data linkage process has been initiated and it is described in figure 1. Data linkage is expected to be completed by 2020. Estimated start and end dates for the project are 1 September 2020 and 1 September 2030, respectively.

*Analysis plan*
Data harmonisation will be performed, and the more reliable data source will be used to guarantee information quality and reliability across exposed and unexposed subjects. To establish the reliability of the source, we will consider how the measurement (eg, self-reported vs measured) and data quality (eg, percentage of missing) were performed. If the same variable (eg, BMI) will be present in the source deemed as most reliable at more than one time point, we will use the measurement closest to the time point of interest.

We will develop a specific statistical analysis plan for each specific study that will be conducted within SOAD. These will follow several general principles. We will aim for the inclusion of all available knee and hip OA and diabetic patients to limit potential selection bias. We will

**Table 3** Description of single variables collected from the SHAR

| SHAR Variable category | Variables | Baseline | Follow-up 1, 6 and 10 years |
|---|---|---|---|
| *Surgery-related variables* | | | |
| Diagnosis (at hip) | ICD-10 | x | |
| | Laterality | x | |
| Date | Date of surgery | x | |
| Type of surgery | Primary, revision, other re-operation | x | |
| Type of replacement | Total, partial, resurfacing hip replacement | x | |
| Patient status | Age, sex, height, weight, ASA class | x | |
| Implant characteristics | Article number, type of implant | x | |
| Technique | Incision, fixation | x | |
| *Patient-reported measures* | | | |
| Smoking status | Smoking (never, ex, daily, not daily) | x | |
| Musculoskeletal comorbidity | Charnley class* | x | x |
| | | x | x |
| Generic | EQ-5D | x | x |
| Treatment before hip replacement surgery | Physiotherapy | x | |
| | Standardised core treatment of education and supervised exercises | x | |
| Disease specific | Hip pain (Likert) | x | x |
| Satisfaction | Satisfaction with treatment (Likert) | | x |

*Charnley class: classifications of musculoskeletal impairment. Class A corresponds to unilateral hip disease; class B indicates bilateral hip disease and class C indicates multiple joint disease or some other condition that inhibits the patient's ability to walk.
ASA, American Society of Anesthesiologists; ICD-10, International Classification of Diseases, tenth revision; SHAR, Swedish Hip Arthroplasty Register.

use multiple imputation methods to impute the missing data on exposures, outcomes and confounders, when relevant. The imputation model will be specific for each study and compatible with the chosen analysis model. For example, the fully conditional specification (also called chained equations) may be used to enable flexible models for proper imputation of all variables. In the statistical modelling we will aim for estimation of causal effects and statistical models will be chosen accordingly using direct acyclic graphs to enable proper confounding control.[31] For confounding control, we will use regression models or inverse probability weighting. For analysis of panel data (ie, longitudinal repeated measurements of the participants and/or data clustered by caregiver) we will use multilevel regression models. For time-to-event data we will use the proportional hazards Cox regression model, or, if appropriate, parametric models. For mediation analysis, we will use linear models or maximum likelihood structural equation models when appropriate. For categorical outcomes we will use other approaches.[32 33] We will report the results from all analyses as relevant estimated effect size (such as risk differences, risk ratios of hazard ratios) with 95% CIs and interpret these for clinical relevance, irrespective of statistical significance.[34 35]

For the current study, we did not perform any power calculation. This because the question of power can be considered secondary in such a setting where the sample size is driven by data availability and not decided a priori. In addition, when interpreting results, we will not use the concept of statistical significance, but we will base our interpretation on effect sizes (and the uncertainty around them) and clinical relevance.[36] Therefore, we will include all the available data, with ~100 000 BOA participants, whereof an estimated 15 000 have diabetes.[37] Regarding the SHAR and SKAR, we will have data for 240 000 and 320 000 joint replacements, respectively. Based on previous studies we expect that the prevalence of diabetes will be around 8% and 14% among patients undergoing THR and TKR, respectively.[38 39] We expect that this will enable precise estimations of the main effects of interest.

## Patient and public involvement

Patient representatives were not involved in the development of the research question or the design of this study. However, patients were involved in the creation of the BOA-supported self-management programme and contributed to the development of the key contents of the programme.[40] Patients are also actively involved in

**Table 4** Description of single variables collected from the SKAR

| SKAR | | Baseline | Follow-up 1 year |
|---|---|---|---|
| **Variable category** | **Variable** | | |
| Surgery-related variables | | | |
| Diagnosis (at knee) | ICD-10 | x | |
| | Laterality | x | |
| Date | Date of surgery | x | |
| Type of surgery | Primary, revision | x | |
| Type of replacement | Total, unicompartmental, stabilised (hinged) knee replacement | x | |
| Patient's status | Age, sex, height, weight, ASA class | x | |
| Implant characteristics | Article number, type of implant | x | |
| Technique | Incision, fixation | x | |
| *Patient-reported measures* | | | |
| Musculoskeletal comorbidity | Charnley class (modified)* | x | x |
| | | x | x |
| Generic | EQ-5D | x | x |
| Satisfaction | Satisfaction with treatment (VAS) | | x |
| Disease specific | KOOS, knee pain (VAS) | x | x |

*Charnley class: classifications of musculoskeletal impairment. Class A: unilateral knee disease; class B1: bilateral OA, one knee is scheduled for or already received arthroplasty surgery while the other knee has OA but no scheduled arthroplasty surgery; B2: bilateral OA, one knee is scheduled for or already received arthroplasty surgery while the other knee has already received knee arthroplasty surgery and class C: multiple joint disease or some other condition that inhibits the patient's ability to walk.
ASA, American Society of Anesthesiologists; ICD-10, International Classification of Diseases, tenth revision; KOOS, Knee injury and Osteoarthritis Outcome Score; VAS, visual analogue scale.

the BOA programme and deliver a compulsory education session where the patient's perspective on OA self-management treatment is explored. The Swedish Hip and Knee Arthroplasty Registers have patient representatives on their respective steering committees.

## ETHICS AND DISSEMINATION
### Storage and management of data
A copy of the full data set will be stored at the Center of Registers Västra Götaland, Gothenburg, Sweden. A second copy of the full data set will be stored at Lund University on the platform LUSEC (Lund information security platform). The platforms are designed to securely store, manage and analyse data in accordance with the European Union general data protection regulation. The process of linkage, storage and management of data, the role of informed consent in register-based research and safeguarding the integrity of study participants follows the legal and ethical frameworks as described by Swedish law and ethical boards. This has been described by Ludvigsson *et al*.[28]

### Dissemination
The results from this study will be published in peer-reviewed scientific journals and will be presented at the leading national and international meetings in the field. The results will also be disseminated through annual reports published on the registers' websites in order to reach clinicians working with people with OA and diabetes.

In order to reach people suffering from OA and diabetes, we aim to exploit the connection between BOA people seeking care for OA. Briefly, we will target BOA educators providing them material (through email and mail) regarding the progress and achievement of the project focusing on the impact that the coexistence of OA and diabetes has on the treatment. In this way, we will reach the new patients taking part in BOA who will be better informed about their condition and about the strategy to undertake in order to maximise the benefit of first-line intervention.

Finally, in SOAD we recognise the importance of reaching a broad audience, and for this reason, we will use the authors' Twitter and Facebook networks to create awareness among the general public of the importance of the issue. The social media will help us to make scientific practice easily accessible and understandable to an audience of non-specialists.

## DISCUSSION
This study cohort will provide unique insights into the relationship between diabetes and OA. By using data from the BOA, SHAR, SKAR and NDR registers, we will be able to investigate the influence of diabetes on the outcome of

**Table 5** Description of single variables collected from the NDR

**NDR**

| Variable category | Variable |
|---|---|
| Patient's characteristics | Age (years), sex, height, weight, BMI |
| Diabetes characteristics | Type of diabetes, HbA1c (mmol/mol), debut year of diabetes, diabetes duration (years), age at onset |
| Diabetes treatment | Diet only, insulin, tablets, tablets and insulin |
| Method of insulin delivery | Insulin Pump Treatment (CSII), MDI |
| Blood pressure | Systolic blood pressure (mm Hg), diastolic blood pressure (mm Hg) |
| Cholesterol | Total cholesterol (mmol/L), LDL (mmol/L), HDL (mmol/L) |
| HbA1c | Triglycerides (mmol/L) |
| Renal function | Creatinine (µmol/L), eGFR (mL/min/1.73 m$^2$) |
| Retinopathy | Retinopathy (yes/no) |
| Other treatments | Anti-hypertensive treatment, lipid-lowering treatment |
| Physical activity | Times per week of moderate to intense physical activity |
| Smoking status | Smoking (yes/no) |
| Albuminuria | Micro-albuminuria, macro-albuminuria (%) |

Variables are measured at least once per year for patients with diabetes type II and four times per year for patients with diabetes type I. If the patient has specific problems, variables may be recorded with higher frequency.
BMI, body mass index;CSII, Continuous Subcutaneous Insulin Infusion; eGFR, estimated glomerular filtration rate; HbA1c, haemoglobin subunit alpha 1c;HDL, high-density lipoprotein; LDL, low-density lipoprotein; MDI, multiple daily injections; NDR, National Diabetes Register.

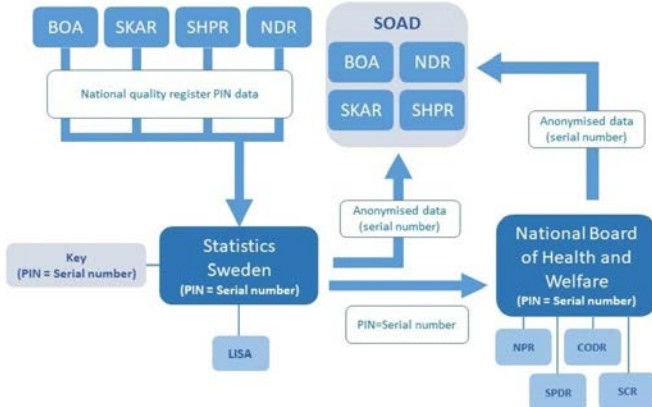

**Figure 1** The data linkage process. Data from the four national quality registers, BOA register, SHAR, SKAR and NDR, are safely transferred to statistics Sweden. Statistics Sweden will anonymise the data by replacing pin with serial numbers. Data will be extracted from LISA (longitudinal integration database for health insurance and labour market studies) and transferred to the entity principally responsible for the SOAD cohort research. The PIN and serial numbers will also be shared with national board of health and welfare which will return data from NPR, SPDR, CODR and SCR to the entity principally responsible for the research. The linkage key will be saved at statistics Sweden for 3 years to allow the possibility of adding more year cohorts or new variables to the research database if new research questions arise (with new ethical approval). BOA, Better management of Patients with osteoarthritis; CODR, Cause of Death Register; NDR, National Diabetes Register; NPR, National Patient Register; PIN, personalidentity number; SCR, Swedish Cancer Register; SHAR, Swedish Hip Arthroplasty Register; SKAR, Swedish Knee Arthroplasty Register; SOAD, Swedish Osteoarthritis and Diabetes; SPDR, Swedish Prescribed Drug Register.

non-surgical and surgical OA interventions as well as on the effect of OA treatments on diabetes control.

To our knowledge, this will be the largest data set combining data on OA and diabetes management. Due to the large sample size on a national level, results arising from this study will likely have high external validity and generalisability. However, treatment data collected in national registers are likely to be influenced by the regional differences in treatment protocols and data collection which characterise different clinical environments when compared with, for example, highly standardised clinical trials.

In conclusion, to optimise treatments for OA and diabetes and move towards a personalised-care approach, it is important to identify factors and comorbidities that may negatively influence the outcome of the interventions. The coexistence of several conditions creates a more complex disease status which requires additional considerations and cares for the patient to experience the desired benefit from the provided interventions. The SOAD cohort will help us to identify these patients with complex needs, opening a venue for the development of better treatment approaches. Ultimately, the cohort has the potential to impact on the way OA is managed when other comorbidities coexist, potentially reducing the huge burden of this disease.

**Author affiliations**
[1]Faculty of Medicine, Department of Clinical Sciences, Orthopedics, Lunds University, Lund, Sweden
[2]Faculty of medicine, Department of Clinical Sciences, Orthopedics, Clinical Epidemiology Unit, Lund University, Lund, Sverige, Sweden
[3]Centre of Registers Västra Götaland, The Swedish Hip Arthroplasty Register, Goteborg, Sweden
[4]Department of Orthopaedics, Sahlgrenska Academy, University of Gothenburg, Institute of Clinical Sciences, Gothenburg, Sweden
[5]National Diabetes Register, Centre of Registers in Region Västra Götaland, Goteborg, Sweden
[6]Department of Molecular and Clinical Medicine, University of Gothenburg, Goteborg, Sweden
[7]National Diabetes Register, Centre of Registers Västra Götaland, Gothenburg, Sweden
[8]Health Metrics Unit, Sahlgrenska Academy, University of Gothenburg, Goteborg, Sweden
[9]Department of Clinical Sciences, Lund University, Lund, Sverige, Sweden

[10] The Swedish Knee Arthroplasty Register, Lund, Sweden
[11] Department of Medical and Health Sciences (IMH), division of physiotherapy, Faculty of Medicine and Health Sciences, Linkoping University, Linköping, Sweden

**Contributors** AD, JV, SL, AMS, AT, SF, EN, AW-D, AA, LD, OR and ME provided substantial contributions to the conception, design of the work and analysis plan of data. All authors contributed to the drafting of the protocol and approved the final version.

**Funding** JV has received funding from Dr. Felix Neuberghs Foundation. ME and AT are funded by the Swedish Research Council and The Swedish Rheumatology Association.

**Competing interests** AW-D is employed at the Swedish Knee Arthroplasty Register (SKAR). JV, A-MS, SF, EN and OR are employed by the Centre of Registers Västra Götaland, Sweden. AW-D is employed at the SKAR. LD is the co-founder and Chief Medical Officer of Joint Academy, a company that provides web-based non-surgical interventions for patients with hip and knee osteoarthritis. AA is employed by the Better Management of OsteoArthritis register.

**Patient consent for publication** Not required.

**Ethics approval** Ethical approval for the creation of the cohort and the analyses as detailed in the present protocol has been obtained (14 May 2019, 2019-02570).

**Provenance and peer review** Not commissioned; externally peer reviewed.

**ORCID iD**
Andrea Dell'Isola http://orcid.org/0000-0002-0319-458X

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
