## [Reviewer comments · BMJ Open]

ARTICLE DETAILS

TITLE (PROVISIONAL)	Understanding the role of diabetes in the osteoarthritis disease and treatment process: a study protocol for The Swedish Osteoarthritis and Diabetes (SOAD) cohort
AUTHORS	Dell'Isola, Andrea; Vinblad, Johanna; Lohmander, Stefan; Svensson, Ann-Marie; Turkiewicz, Aleksandra; Franzén, Stefan; Naclér, Emma; W-Dahl, A; Abbott, Allan; Dahlberg, L; Rolfson, Ola; Englund, Martin

VERSION 1 - REVIEW

REVIEWER	Nina Østerås National Advisory Unit on Rehabilitation in Rheumatology, Diakonhjemmet Hospital, Norway
REVIEW RETURNED	20-Sep-2019

GENERAL COMMENTS	General: The manuscript is a protocol for a planned (potentially ongoing?) cohort study on register data from an osteoarthritis (OA) non-surgical intervention register, two hip and knee arthroplasty registers and a diabetes register. Additional data will be extracted from a drug prescription register, the National Patient Register, a cancer register and a cause of death register as well as demographic information from Statistics Sweden. The study aims to increase knowledge on the influence that diabetes has on the outcomes of OA patients who have received non-surgical and/or surgical treatments for hip and knee OA, and the influence OA and its treatments have on the diabetes control. The study is important as it will exploit the unique potential in merging data from the many national registers in Sweden to target an area of limited knowledge among those with a coexistence of OA and diabetes. The study is likely to be of interest for researchers, clinicians and patients worldwide. The protocol is well written and presented, but a few aspects may increase the readability for the readers. Major comments: It is not 100% clear whether the authors plan to focus on hip and knee OA only. The BOA intervention and register include people with hip/knee pain, but some may also have hand OA or OA in more joints. The two arthroplasty registers include data on surgery in the hip and knee joints. From Table 1 I understand that BOA participants may indicate hand as their most affected joint. Since
--

the relationship between diabetes and hand OA (non-weight bearing joints) is more debated and may be somewhat different from the relationship between diabetes and hip/knee OA, I think the authors should clearly define early in the protocol whether they focus on hip/knee or if they include OA in all joints. This should also be reflected in the Introduction, and in particularly in the Analysis plan on page 10, line 218.

Minor comments:

1. Page 4, Article summary: the acronyms BOA, SHPR and SKAR needs to be expanded/explained.
2. Page 4, Article summary: The sentence "Certain treatments are given to patients with specific characteristics..." is unclear. Either you should provide an example of such certain treatment and specific characteristics, or you could consider rewriting this sentence and write that due to the complexity of the OA disease, the treatment is individualized... if that is what you meant.
3. Page 6, line 98: Weight loss or weight control is also included in first-line management for hip and knee OA. Consider to mention this. If hand OA is also included, then there are more first-line treatment modalities that are recommended...
4. Page 7, research question 7: The effect of non-surgical treatment on may have a different effect on diabetes control for those with type 2 diabetes as compared to type 1. Do you plan to do separate analyses for the two types of diabetes? Consider to be more specific on this and also whether this should be mentioned in the introduction section.
5. Page 7, research question 7: At what time points to you plan to do these analyses? Compare diabetes control before-after the non-surgical intervention? According to Table 4, those with type 2 diabetes are only measured once per year...
6. Page 7, line 147: The table first mentioned in the text should be named Table 1. Tables should be cited in ascending numerical order in the main text of the article.
7. Page 8, line 176-7: I believe the BOA also offers a third session with a patient representative focusing on self-management. Consider to include this information here, and also in the PPI section on page 10.
8. Page 9, lines 206-12: The authors provide no information on the time point for when the linkage of the register data will be/has been performed. Consider to include information on this.
9. Page 10, line 235: The authors state that data for nearly 30,000 joint replacements will be included.... I do not understand how you got to this number. The text on the SHAR and SKAR mentions far larger number of joint replacements...? Also, the expected number of individuals registered in NDR is not mentioned in this paragraph.
10. Page 11, Ethics and dissemination: The abstract contains information on having received ethical approval for this study. This information should also be included and expanded upon in the Ethics section of the main text
11. Page 11, lines 254-7: The results of this study are likely to be of interest, not only among clinicians and researchers, but also among people suffering from OA and diabetes. Hence, plans for dissemination of lay information should also be included in this paragraph, which highlights that patient research partners should be included in the project group, or at least engaged in the study. The lack of PPI involvement is a minor limitation of this study.
12. Table 1: Explanations for c) and d) are missing in the footnote.

	13. Table 5: In the column to the far right, “disease progression” is mentioned as an example confounder or effect modifier. Does this relate to diabetes or OA? 14. Table 5: As weight loss intervention represents first-line treatment of OA, and also is important for diabetes type 2, some of the patients included in this study may have participated in such interventions. I understand that the BOA does not include this and that information of participation in such interventions will not be available, but do you plan to include not only BMI, but also changes in weight as a potential cofounder or effect modifier? 15. Table 5: In the diabetes population being exposed to non-surgical OA treatment, I think “Type of diabetes” should be considered included as a potential confounder/effect modifier. 16. Page 25, CONSORT checklist, item#10 study size: The authors refer to Figure 1, but this explains the linkage and does not provide any numbers. Include also reference to the main text regarding information on the registers and the paragraph on sample size.
--	--

REVIEWER	Rebecca J. Cleveland Thurston Arthritis Research Center University of North Carolina at Chapel Hill USA
REVIEW RETURNED	04-Oct-2019

GENERAL COMMENTS	This nicely written manuscript describes the proposed creation of a population-based cohort that will result from the combining of 4 national sources of diabetes, knee and hip OA management and replacement, as well as 5 other national general registries. However, It is unclear whether any of this has taken place yet or if it is still in the planning phase. The authors detailed nicely when the separate registers were started, but there is no mention of a timeline for the combined cohort. Is a start date scheduled? Also there is little detail on how the 4 registers and other national registries will be combined other than it will be matched by the Swedish PIN that all residents are assigned. I imagine there will be a great number of challenges in combining nine different registries. Are there plans in place for dealing with these challenges? Are there plans in place for harmonizing the variables that are collected in different manners? While described as a “protocol” it is somewhat light on details about how some of the procedures are going to be carried out. It has a lot of descriptions of the cohorts themselves which is nice, but it would be helpful to understand how they plan on putting all these cohorts together and methods to make compatible, usable data. This study has the potential to answer a great number of highly important and timely questions. It will be a great resource for many different types of investigations and I look forward to seeing the research that will come out of it. There are just some clarifications I’d like to see. Specific comments: Methods and Analysis Page 6, line 118: If you think there are similar processes affecting disease progression in both conditions, what are the plans to disentangle what is going on? If there are earlier manifestations of
--

	one disease before the other manifests itself does it necessarily mean that one occurred before the other? What is the natural course of disease in both conditions? Page 6, line 123: Here you are referring to cross-sectional associations, yes? Page 7, line 128: What are these OA self-management measures? It is not clear from the text or Table 1. Page 7, line 131: Is this proposed to be a longitudinal analysis? Do you have requirements for length of follow-up? Page 7, line 132: How do you know people are doing their self-management using register data? Are they being tracked over time? Page 7, line 139: Is this longitudinal or cross-sectional? Page 7, line 147: Why is Table 5 first? Page 8, line 176: I am not familiar with the BOA program so a little more detail here would help. Everyone get the two education sessions, correct? What percentage of the BOA participants actually do one or both? After the initial education sessions, are they allowed to participate in one or both of the additional programs? For instance, can you get both the one-time individual session with a physiotherapist AND 12 group sessions? What are the participation rates for both of those programs? How many of the group sessions does a participant typically attend? It might help to have the protocol for the BOA program attached as an Appendix. Page 8, line 179: Is this 12 sessions total, or 12 per year? Page 8, line 183: Do they also get a questionnaire at the beginning? If so, it should be mentioned that they do have a baseline measurement of these factors. Page 9, line 200: How is the completeness of these registries assessed? Page 10, line 213: Has this ethics approval been obtained? Or are you speaking of the separate health registries? Page 10, line 216:  • I understand why there was no power calculation since you don't know how many people you will have, but there should be some sort of basic sample size calculation to give the reader an idea of what sort of sample size will be needed to detect at least your main effects. • Also, what types of associations are you expecting to find, what range of strengths of association? Page 10, line 217: Is there any plan to harmonize the data? It seems like there will be a lot of variables that are not exactly the same, so something will need to be done to ensure comparability. Discussion: Page 11, line 258: I have some concerns about being able to merge so many datasets that will provide usable data since they come from different sources and have different measures. Can you comment on what measures are being taken to ensure this occurs and what protocols are in place if challenges arise?
--	--

VERSION 1 – AUTHOR RESPONSE

Reviewer: 1

Reviewer Name: Nina Østerås

Reviewer comment:

General:

The manuscript is a protocol for a planned (potentially ongoing?) cohort study on register data from an osteoarthritis (OA) non-surgical intervention register, two hip and knee arthroplasty registers and a diabetes register. Additional data will be extracted from a drug prescription register, the National Patient Register, a cancer register, and a cause of death register as well as demographic information from Statistics Sweden. The study aims to increase knowledge on the influence that diabetes has on the outcomes of OA patients who have received non-surgical and/or surgical treatments for hip and knee OA, and the influence OA and its treatments have on the diabetes control.

The study is important as it will exploit the unique potential in merging data from the many national registers in Sweden to target an area of limited knowledge among those with a coexistence of OA and diabetes. The study is likely to be of interest for researchers, clinicians and patients worldwide. The protocol is well written and presented, but a few aspects may increase the readability for the readers.

Major comments:

It is not 100% clear whether the authors plan to focus on hip and knee OA only. The BOA intervention and register include people with hip/knee pain, but some may also have hand OA or OA in more joints. The two arthroplasty registers include data on surgery in the hip and knee joints. From Table 1 I understand that BOA participants may indicate hand as their most affected joint. Since the relationship between diabetes and hand OA (non-weight bearing joints) is more debated and may be somewhat different from the relationship between diabetes and hip/knee OA, I think the authors should clearly define early in the protocol whether they focus on hip/knee or if they include OA in all joints. This should also be reflected in the Introduction, and in particularly in the Analysis plan on page 10, line 218.

Author response:

We thank the reviewer for the comment, and we agree that it is important to specify the focus of the study considering the differences between hip/knee OA and hand OA. Our plan is to focus only on knee and hip OA. We modified the manuscript to better reflect this aspect. We also modified the aim of the study to clarify that we will focus on hip and knee OA.

Author action:

Thorough the introduction we added knee and hip before OA when necessary to increase clarity.

Line 119: This study cohort is created to increase knowledge of the influence that diabetes has on the outcomes of OA patients who have received non-surgical and/or surgical treatments for hip and knee OA, and the influence that hip and knee OA and its treatments have on the diabetes control.

Reviewer comment:

1. Page 4, Article summary: the acronyms BOA, SHPR and SKAR needs to be expanded/explained.

Authors response:

We apologise for the mistake, we provided clarification for the acronyms

Author action:

Line 67: People included in the National Quality Register for Better Management of Patients with Osteoarthritis (BOA) Register, the Swedish Hip Arthroplasty Register (SHAR), the Swedish Knee Arthroplasty Register (SKAR) received an intervention due to OA. Certain treatments are given to patients with specific characteristics, which implies that selection bias and confounding by indication may bias our estimates.

Reviewer comment:

2. Page 4, Article summary: The sentence "Certain treatments are given to patients with specific characteristics..." is unclear. Either you should provide an example of such certain treatment and specific characteristics, or you could consider rewriting this sentence and write that due to the complexity of the OA disease, the treatment is individualized... if that is what you meant.

Authors response:

We would like to thank the reviewer for the suggestion, we agree that more clarity is needed. What we meant was that, as correctly stated, treatments in primary cares are adapted and individualised which need to be considered when interpreting results from this cohort.

Author action:

Line 70: Due to the complexity of the OA disease, treatments are individualised based on patient's and disease characteristics, which implies that selection bias and confounding by indication may bias our estimates.

Reviewer comment:

3. Page 6, line 98: Weight loss or weight control is also included in first-line management for hip and knee OA. Consider to mention this. If hand OA is also included, then there are more first-line treatment modalities that are recommended...

Authors response:

We would like to thank the reviewer for the suggestion, we added weight loss as part of the first-line intervention for people overweight. We did not add other interventions regarding hand OA since our focus will be exclusively on hip and knee OA.

Author action:

Line 103: The evidence-based first-line management for people with hip and knee OA includes education and exercise, which are recommended regardless of OA disease severity, and weight loss for those overweight [19].

Reviewer comment:

4. Page 7, research question 7: The effect of non-surgical treatment on may have a different effect on diabetes control for those with type 2 diabetes as compared to type 1. Do you plan to do separate analyses for the two types of diabetes? Consider to be more specific on this and also whether this should be mentioned in the introduction section.

Authors response:

We fully agree that this is an important point and we will take it into consideration either by stratifying the analysis or adjusting the estimates. We included type of diabetes as a potential confounder or effect modifier in table 1. However, we also made the following changes where the main aims of the study are described to clarify that we will take into account type of diabetes:

Author action:

Line 126: The research questions cover two main thematic areas: (1) the impact of diabetes on the outcome of non-surgical and surgical OA treatments, and (2) the impact of OA non-surgical and surgical OA treatments on diabetes control (consideration to type of diabetes (I or II) will be taken).

Reviewer comment:

5. Page 7, research question 7: At what time points do you plan to do these analyses? Compare diabetes control before-after the non-surgical intervention? According to Table 4, those with type 2 diabetes are only measured once per year...

Authors response:

As correctly stated, we aim to measure diabetes control before and after the intervention to measure whether any change in diabetes control will happen during the period of intervention. Persons with type 2 diabetes are measured at least once a year. In case of complications for type 2 diabetes and for type 1 diabetes in general more frequent measures will be available. We plan to use all the available data.

Reviewer comment:

6. Page 7, line 147: The table first mentioned in the text should be named Table 1. Tables should be cited in ascending numerical order in the main text of the article.

Authors response:

we would like to thank the reviewer for spotting this mistake, we re-numbered the tables accordingly to their order of appearance in the manuscript

Reviewer comment:

7. Page 8, line 176-7: I believe the BOA also offers a third session with a patient representative focusing on self-management. Consider to include this information here, and also in the PPI section on page 10.

Authors response:

We thank the reviewer for the suggestions. BOA does include a third session with a patient-representative, however, this is not mandatory and it is not offered in all the clinics. However, we decided to include this information in the manuscript as suggested.

Author action:

Line 187: A third, optional, session held by a trained osteoarthritis communicator (a patient with osteoarthritis who previously participated in BOA) is offered to provide a patient perspective on OA self-management and to teach about the lived experience with this condition, as well as his or her personal experience of non-surgical interventions.

Reviewer comment:

8. Page 9, lines 206-12: The authors provide no information on the time point for when the linkage of the register data will be/has been performed. Consider to include information on this.

Authors response:

We agree with the reviewer that this information is needed, we decided to make the following changes:

Author action:

Line 229: Data linkage for the current study will include all the data available in BOA, SHPR, SKAR and NDR and it will start from the first time point available in the registers. The data linkage process has been initiated and it is described in figure 1. Data linkage is expected to be completed by 2020.

Reviewer comment:

9. Page 10, line 235: The authors state that data for nearly 30,000 joint replacements will be included.... I do not understand how you got to this number. The text on the SHAR and SKAR mentions far larger number of joint replacements...? Also, the expected number of individuals registered in NDR is not mentioned in this paragraph.

Authors response:

We would like to apologise for the mistake, we amended the sentence as follow:

Author action:

Line 265: Regarding the SHAR and SKAR, we will have data for 240 000 and 320 000 joint replacements respectively. Based on previous studies we expect that the prevalence of diabetes will be around 8% and 14% among patients undergoing THR and TKR, respectively [37, 38]. We expect that this will enable precise estimation of the main effects of interest.

Reviewer comment:

10. Page 11, Ethics and dissemination: The abstract contains information on having received ethical approval for this study. This information should also be included and expanded upon in the Ethics section of the main text

Author action:

Line 279: An ethical approval for the creation of the cohort and the analysis detailed in this protocol has been obtained (14th May 2019, 2019-02570).

Reviewer comment:

11. Page 11, lines 254-7: The results of this study are likely to be of interest, not only among clinicians and researchers, but also among people suffering from OA and diabetes. Hence, plans for dissemination of lay information should also be included in this paragraph, which highlights that patient research partners should be included in the project group, or at least engaged in the study. The lack of PPI involvement is a minor limitation of this study.

Authors response:

We thank the reviewer and we agree on the importance of dissemination among the general public and especially among people suffering from OA and diabetes. For this reason, we decided to design a more detailed outreaching strategy that will comprise not only traditional means of communication but will exploit social media and the connection between BOA and people seeking care for OA. We made the following changes to better describe our outreaching strategy.

Author action:

Line 290: The results from this study will be published in peer-reviewed scientific journals and will be presented at the leading national and international meetings in the field. The results will also be disseminated through annual reports published on the registers' websites in order to reach clinicians working with people with OA and diabetes.

In order to reach people suffering from OA and diabetes, we aim to exploit the connection between BOA people seeking care for OA. Briefly, we will target BOA educators providing them material (through email and mail) regarding the progress and achievement of the project focusing on the impact that the coexistence of OA and diabetes has on the treatment. In this way, we will reach the new patients taking part in BOA that will be more informed about their condition and about the strategy to undertake to obtain the best results from a first-line intervention.

Finally, In SOAD we recognize the importance of reaching a broad audience, for this reason, we will utilise the authors Twitter and Facebook networks to create awareness among the general public of the importance of the issue. The social media will help us to make scientific practice easily accessible and understandable to an audience of non-specialists.

Reviewer comment:

12. Table 1: Explanations for c) and d) are missing in the footnote.

Authors response:

Thank you for noticing the missing footnotes. We provided clarification as suggested

Author action:

Line 458: c) Charnley class: classifications of musculoskeletal impairment. Class A corresponds with unilateral hip or knee OA; class B bilateral hip or knee OA and class C indicates multiple joint OA or some other condition that inhibits the patient's ability to walk

d) Answering to: "Select the box that corresponds to your average pain from your most affected joint the last week".

Reviewer comment:

13. Table 5: In the column to the far right, "disease progression" is mentioned as an example confounder or effect modifier. Does this relate to diabetes or OA?

Authors response:

In the table, we refer to diabetes disease progression. However, with disease progression we wanted to refer to the duration of the disease. Therefore we decided to change the wording to better convey the message:

Author action:

Table 1: Diabetes disease duration (age at diagnosis, duration of disease)

Reviewer comment:

14. Table 5: As weight loss intervention represents first-line treatment of OA, and also is important for diabetes type 2, some of the patients included in this study may have participated in such interventions. I understand that the BOA does not include this and that information of participation in such interventions will not be available, but do you plan to include not only BMI, but also changes in weight as a potential cofounder or effect modifier?

Authors response:

we agree with the reviewer about the importance of weight change. We will consider including weight change as confounders. We, therefore, added weight change in table 1 as potential confounder or treatment modifier.

Reviewer comment:

15. Table 5: In the diabetes population being exposed to non-surgical OA treatment, I think "Type of diabetes" should be considered included as a potential confounder/effect modifier.

Authors response:

We added type of diabetes as suggested

Reviewer comment:

16. Page 25, CONSORT checklist, item#10 study size: The authors refer to Figure 1, but this explains the linkage and does not provide any numbers. Include also reference to the main text regarding information on the registers and the paragraph on sample size.

Authors response:

We thank the reviewer for the suggestion. We revised the checklist adding the page number where we discuss the number of people within each register and the number of people with OA and diabetes we expect to have in the dataset and the reasons for not including a sample size calculation.

Authors action:

The STROBE checklist has been modified.

Reviewer: 2

Reviewer Name: Rebecca J. Cleveland

Reviewer comment:

This nicely written manuscript describes the proposed creation of a population-based cohort that will result from the combining of 4 national sources of diabetes, knee and hip OA management and replacement, as well as 5 other national general registries. However, It is unclear whether any of this has taken place yet or if it is still in the planning phase. The authors detailed nicely when the separate registers were started, but there is no mention of a timeline for the combined cohort. Is a start date scheduled? Also there is little detail on how the 4 registers and other national registries will be combined other than it will be matched by the Swedish PIN that all residents are assigned. I imagine there will be a great number of challenges in combining nine different registries. Are there plans in place for dealing with these challenges? Are there plans in place for harmonizing the variables that are collected in different manners? While described as a "protocol" it is somewhat light on details about how some of the procedures are going to be carried out. It has a lot of descriptions of the cohorts themselves which is nice, but it would be helpful to understand how they plan on putting all these cohorts together and methods to make compatible, usable data.

This study has the potential to answer a great number of highly important and timely questions. It will be a great resource for many different types of investigations and I look forward to seeing the research that will come out of it. There are just some clarifications I'd like to see.

Reviewer comment:

Methods and Analysis

Page 6, line 118: If you think there are similar processes affecting disease progression in both conditions, what are the plans to disentangle what is going on? If there are earlier manifestations of one disease before the other manifests itself does it necessarily mean that one occurred before the other? What is the natural course of disease in both conditions?

Authors response:

We thank the reviewer for the comment and we agree about the importance of carefully analysing the influence of shared risk factors and causal mechanisms influence the outcomes in analysis. For this reason we will adjust all the analysis for known confounders in order to minimize the bias possibly arising from the complex relationship between OA and diabetes. This will be done as described in the analysis paragraph and it will depend on the specific research question in analysis. Despite this, we would like to highlight that we will not be able to established causality due to the observational nature of the data included in the cohort. In fact, patients are included in one of the registers when they already have the condition and are receiving treatments for that condition. Therefore, we will not have the possibility to answer research questions focusing on the natural course of the diseases.

Reviewer comment:

Page 6, line 123: Here you are referring to cross-sectional associations, yes?

Authors response:

Yes, we refer to the prevalence which we will measure cross-sectionally at the time of enrollment.

Reviewer comment:

Page 7, line 128: What are these OA self-management measures? It is not clear from the text or Table 1.

Authors response:

We thank the reviewer for the comment. The self-management intervention will be the treatment provided in BOA and described in the methods. The outcome measures of the self-management intervention (BOA) are change in pain, pain frequency, and walking difficulties as reported in brackets.

Reviewer comment:

Page 7, line 131: Is this proposed to be a longitudinal analysis? Do you have requirements for length of follow-up?

Authors response:

This will be a longitudinal analysis. We don't have any requirements for length of follow-up. However, considering that BOA started in 2008 the longest time to follow-up for surgery will be 11 years since we expect to receive complete data from the registers up to January 2020. We will consider further updates of the data in approximately 5 years to add further patient numbers, follow-up time, and outcome events.

Reviewer comment:

Page 7, line 132: How do you know people are doing their self-management using register data? Are they being tracked over time?

Authors response:

We thank the reviewer for the comment. Unfortunately, people are not tracked over time and we have data regarding attended sessions only for people undergoing supervised exercise. This means that it is not possible to control whether people are performing home exercise or not. However, we have a self-reported measure of how often the person uses the self-management advice received in BOA. However, this question provides only limited information on the self-management of the symptoms and on the home exercise adherence.

Reviewer comment:

Page 7, line 139: Is this longitudinal or cross-sectional?

Authors response:

Yes, this is intended as a longitudinal analysis. We modified the current research question to better reflect the design of the intended analysis.

Author action:

Line 147: How does a self-management non-surgical intervention for OA influence diabetes (type I vs type II) control (e.g. change in diabetes drug intake after the intervention, change in Hb A1c levels after the intervention) compared to comparable people with diabetes who had not taken part in the intervention?

Reviewer comment:

Page 7, line 147: Why is Table 5 first?

Authors response:

We apologise for the mistake, we have re-numbered the tables according to order of appearance.

Reviewer comment:

Page 8, line 176: I am not familiar with the BOA program so a little more detail here would help. Everyone get the two education sessions, correct? What percentage of the BOA participants actually do one or both? After the initial education sessions, are they allowed to participate in one or both of the additional programs? For instance, can you get both the one-time individual session with a physiotherapist AND 12 group sessions? What are the participation rates for both of those programs? How many of the group sessions does a participant typically attend? It might help to have the protocol for the BOA program attached as an Appendix.

Authors response:

We thank the reviewer for the suggestion, and we agree that more details regarding the treatment may help readers who are not familiar with the BOA programme. All participants in BOA need to attend both education sessions in order to be allowed in the second part of the programme. After the education, all patients who decide to continue with the exercise part of the programme receive a 1 to 1 session with a physiotherapist. This session is attended regardless of whether the participants decide to exercise at home or to attend the supervised group session. All the participants are free to decide how to exercise. Regarding the percentage of people who attended home or supervised exercise, roughly 60% of the people participating in the exercise decided to do so in the supervised group sessions. Roughly 60% of the people doing supervised exercise attended 10 or more sessions.

Author action:

Line 185: BOA offers all the patients two education sessions focusing on the pathophysiology of OA and the benefit of exercise which are mandatory for participating in second (exercise) part of the programme. A third, optional, session held by a trained osteoarthritis communicator (a patient with osteoarthritis who previously participated in BOA) is offered to provide a patient's perspective on OA self-management and to teach about the lived experience with this condition, as well as his or her personal experience of non-surgical interventions. After the education, participants can take part in the exercise phase of BOA which consists of a face-to-face session with a physiotherapist. In this session, the patients receive a personalised intervention programme and the necessary instructions to perform it independently at home. thereafter, participants are given the possibility to perform their exercise programme on their own or to participate in up to 12 supervised group exercise session with a physiotherapist provided two times a week for six weeks.

Reviewer comment:

Page 8, line 179: Is this 12 sessions total, or 12 per year?

Answer: 12 sessions are part of the BOA programme and are provided twice per week over a period of six weeks. Please see above for the clarification made to the manuscript.

Reviewer comment:

Page 8, line 183: Do they also get a questionnaire at the beginning? If so, it should be mentioned that they do have a baseline measurement of these factors.

Authors response:

We would like to thank the reviewer for the suggestion. As correctly noticed the participants complete a questionnaire also before the intervention. We have modified the sentence accordingly.

Author action:

Line 197: The register has a data completeness of almost 90% and the BOA participants have answered validated and patient-relevant socio-demographic and outcome questionnaires at baseline, after the interventions (2-5 months) and at one year (12-15 months) (Table 2).

Reviewer comment:

Page 9, line 200: How is the completeness of these registries assessed?

Authors response:

Completeness of procedures is assessed differently in the different registers. The SKAR and SHAR completeness is estimated by comparing the arthroplasties in the registers to admissions in the

National patient register (NPR) and assuming that the true number of admissions is the combined number of admissions in both registers. The NPR is performing the completeness analysis for both SKAR and SHAR.

Because the NPR does not include data from primary health care, completeness of BOA data is assessed by cross-checking with available health care region databases comparing the records in BOA to the prevalence of the Swedish National Classification of Health Interventions code GB020 (which includes: Information and education about osteoarthritis, standardized and evidence based fitness, strength, activity and function training).

Considering the specificity of this procedure with the Swedish health care system we thought that it would have been better not to include the information in the manuscript which may result hard to understand for readers not familiar with the Swedish registers system.

Reviewer comment:

Page 10, line 213: Has this ethics approval been obtained? Or are you speaking of the separate health registries?

We thank the reviewer for the comment. In the line mentioned we meant that ethical approval is necessary for every project aiming to perform a data linkage. We have obtained ethical approval for the creation of SOAD, however, we noticed that this was not clear for the manuscript. We made the following changes

Author action:

Line 280: An ethical approval for the creation of the cohort, and the analyses as detailed in the present protocol has been obtained (14th May 2019, 2019-02570).

Reviewer comment:

Page 10, line 216:

- I understand why there was no power calculation since you don't know how many people you will have, but there should be some sort of basic sample size calculation to give the reader an idea of what sort of sample size will be needed to detect at least your main effects.
- Also, what types of associations are you expecting to find, what range of strengths of association?

Authors response:

We appreciate that the reviewer understands the difficulty behind performing power calculations in our situation. In addition, different power calculations would be necessary for different research questions. However, we fully understand the reviewer's point of view, and we suggest that for a time to event analysis (e.g. change in risk of receiving a joint replacement) a power calculation may be particularly useful. In a very simple approach, 631 events would be needed assuming power of 80% and a 25% hazard increase (or a 25% hazard decrease (e.g. for incidence of joint replacement)). Given the expected sample size we expect that the observed number of events will largely exceed 631.

However, we would like to refrain from including examples like the one above in the manuscript because they are too generic. Also, our database will be one of the biggest possible with a unique collection of relevant variables and thus we believe will provide useful information. Further, the question of power is secondary in such a setting (sample size calculation is used to decide on sample size, while in our case the sample size is driven by data availability) and we will not use the concept of statistical significance when interpreting the results (Wasserstein RL, Schirm AL, Lazar NA. Moving to a World Beyond "p < 0.05". *The American Statistician*. 2019;73(sup1):1-19.). Instead, we will interpret the estimated effect sizes (and the uncertainty around them) and their clinical relevance. Despite this, we agree with the reviewer that clarifications regarding power calculation should be given to the reader. We decided to amend the manuscript explaining the reason for not providing a power calculation.

Author action:

Line 259: For the current study, we did not perform any power calculation. This because the question of power can be considered secondary in such a setting where the sample size is driven by data availability and not decided a priori. In addition to this, when interpreting results, we will not use the concept of statistical significance, but we will base our interpretation on effect sizes (and the uncertainty around them) and clinical relevance [36].

Reviewer comment:

Page 10, line 217: Is there any plan to harmonize the data? It seems like there will be a lot of variables that are not exactly the same, so something will need to be done to ensure comparability.

Authors response:

We agree with the reviewer that data harmonisation may become an important issue that will be considered when the dataset becomes available. However, the purpose of using data from different registers is that these sources will provide specific data not included in other registers. For example, data on diabetes will come only from the diabetes register, as well as data on joint replacement will come only from the replacement registers. This will reduce the need for data harmonization.

However, there will be a number of variables that will be present on more than one register. We will perform harmonisation using the more reliable data source to guarantee information quality and reliability across exposed and unexposed subjects. Nevertheless, many variables like BMI and NRS pain have common definition that facilitates merging and harmonization. In addition, we will have collection dates for the variables. This will help us choose the most appropriate measure (e.g. for confounding control we would select the BMI values measured closest to the exposure and outcome). If in the source deemed most reliable the same variables will be present at multiple time points we will

use the entry closest to the time point of interest. We decide to add the following sentence in the manuscript to address this issue:

Line 236: Data harmonisation will be performed, the more reliable data source will be used to guarantee information quality and reliability across exposed and unexposed subjects. To establish reliability of the source we will consider how the measurement was performed (e.g self-reported vs measured) and data quality (e.g. percentage of missing). If the same variable (e.g. BMI) will be present in the source deemed as most reliable at more than one time-point, we will use the measurement closest to the time point of interest.

Reviewer comment:

Page 11, line 258: I have some concerns about being able to merge so many datasets that will provide usable data since they come from different sources and have different measures. Can you comment on what measures are being taken to ensure this occurs and what protocols are in place if challenges arise?

Authors response:

We thank the reviewer for the comment. Generally, the need for including all the registers is that they provide unique information on the specific disease/procedure that they are devoted to, while they are all based on unique personal identification number enabling efficient merging. Our experience from similar large register-based studies suggest that a “data conflict” is unlikely. Also, for each study question, specific variables at specific time points relevant to the study question will be retrieved.

VERSION 2 – REVIEW

REVIEWER	Nina Østerås National Advisory Unit on Rehabilitation in Rheumatology, Department of Rheumatology, Diakonhjemmet Hospital, Norway
REVIEW RETURNED	10-Nov-2019

GENERAL COMMENTS	The authors have responded satisfactorily on all my comments. I have no further comments.
---

REVIEWER	Rebecca J. Cleveland Thurston Arthritis Research Center Department of Medicine University of North Carolina at Chapel Hill Chapel Hill, NC USA
REVIEW RETURNED	26-Nov-2019

GENERAL COMMENTS	The authors clearly put a lot of work into the revision and their thoughtfulness is appreciated. I look forward to seeing the research results from this study protocol.
--